# Accuracy of rapid lateral flow immunoassays for human leptospirosis diagnosis: A systematic review and meta-analysis

Teerapat Nualnoi [1,2] *, Luelak Lomlim [3,4], Supawadee Naorungroj [5] *

**1** Department of Pharmaceutical Technology, Faculty of Pharmaceutical Sciences, Prince of Songkla University, Hat Yai, Songkhla, Thailand, **2** Drug Delivery System Excellence Center (DDSEC), Faculty of Pharmaceutical Sciences, Prince of Songkla University, Hat Yai, Songkhla, Thailand, **3** Department of Pharmaceutical Chemistry, Faculty of Pharmaceutical Sciences, Prince of Songkla University, Hat Yai, Songkhla, Thailand, **4** Phytomedicine and Pharmaceutical Biotechnology Excellence Center (PPBEC), Faculty of Pharmaceutical Sciences, Prince of Songkla University, Hat Yai, Songkhla, Thailand, **5** Department of Conservative Dentistry, Faculty of Dentistry, Prince of Songkla University, Hat Yai, Songkhla, Thailand

\* teerapat.n@psu.ac.th (TN); supawadee.n@psu.ac.th (SN)

## Abstract

### Background

In the last two decades, several rapid lateral flow immunoassays (LFIs) for the diagnosis of human leptospirosis were developed and commercialized. However, the accuracy and reliability of these LFIs are not well understood. In this study, we aimed to evaluate the accuracy of leptospirosis LFIs as well as the factors affecting the test efficiency using systematic review and meta-analysis.

### Methods and results

Original articles reporting the accuracy of human leptospirosis LFIs against microagglutination tests (MAT) or immunofluorescent assays (IFA) were searched from PubMed, Embase, and Scopus, and selected as per pre-set inclusion and exclusion criteria. A total of 49 data entries extracted from 24 eligible records published between 2003 and 2023 were included for meta-analysis. A meta-analysis was performed using STATA. The quality of the included studies was assessed according to the revised QUADAS-2. Only nine studies (32.1%) were considered to have a low risk of bias and no concern for applicability. Pooled sensitivity and specificity were calculated to be 68% (95% confidence interval, CI: 57–78) and 93% (95% CI: 90–95), respectively. However, the ranges of sensitivity (3.6 – 100%) and specificity (53.5 – 100%) of individual entries are dramatically broad, possibly due to the heterogeneity found in both study designs and LFIs themselves. Subgroup analysis demonstrated that IgM detection has better sensitivity than detection of IgG alone. Moreover, the test performance seems to be unaffected by samples from different phases of infection.

**Data Availability Statement:** All relevant data are within the manuscript and its Supporting Information files. The raw data used to generate plots/graphs presenting the pooled sensitivity and

specificity values are mainly presented in Table 3. Additionally, the calculated values have been reported along with their 95% confidence interval.

**Funding:** TN was supported by a Prince of Songkla University Research Grant (grant number: PHA6202066S) to conduct this research. The funder had no role in study design, data collection and analysis, decision to publish, or preparation of the manuscript.

**Competing interests:** The authors have declared that no competing interests exist.

## Conclusions

The pooled specificity of LFIs observed is somewhat acceptable, but the pooled sensitivity is low. These results, however, must be interpreted with caution because of substantial heterogeneity. Further evaluations of the LFIs with well-standardized design and reference test will be needed for a greater understanding of the test performance. Additionally, IgM detection type should be employed when leptospirosis LFIs are developed in the future.

## Author summary

Several rapid lateral flow immunoassays (LFIs) for the diagnosis of leptospirosis, the most common bacterial infection transmitted from animals to humans, have been developed during the last two decades. The test accuracy, however, seems inconsistent among studies, raising questions about their reliability and applicability. Systematic review and meta-analysis were exploited to answer these questions. A total of 28 studies evaluating the human leptospirosis LFIs were included in this review. Major findings from our analysis include: i) the overall specificity of the LFIs is likely acceptable (93%), but the sensitivity is significantly low (68%); ii) substantial variation among studies was observed, alerting the reliability of this meta-analysis results; iii) the accuracy of the LFIs seems to be unaffected by samples collected from different phases of infection (acute vs. convalescent); and iv) IgM detection LFIs exhibit higher sensitivity as compared to IgG detection type. From our findings, we suggest that IgM detection LFIs should be focused on for future development. Also, in order to understand the performance of the LFIs better, other evaluation studies are still needed.

## Introduction

Leptospirosis is the most prevalent zoonotic infections worldwide, with the highest incidence occurring in tropical resource-poor nations [1]. It was estimated to be responsible for 1.03 million cases and 58,900 deaths annually [2]. Humans may acquire the infection from direct contact with infected animals' fluids or from water or soil contaminated with those through cuts, abrasive skin, or mucosal contact [3]. Leptospirosis cases are usually asymptomatic, but some patients may develop symptoms. The most common symptom observed among leptospirosis patients is an acute febrile illness. The infection can be treated by using common antibiotics such as doxycycline, and azithromycin. However, if the appropriate treatment is not given in a timely manner, the disease may progress to more severe clinical manifestations, which could potentially lead to multiple organ failure and death [1]. Early and accurate diagnosis, therefore, plays a key role in the management of the infection. Unfortunately, clinical presentations of leptospirosis are unspecific, making the disease difficult to differentiate from other infections, e.g., malaria and dengue.

Leptospirosis is caused by spirochete bacteria belonging to the genus *Leptospira*. More than 300 serovars of *Leptospira* spp. have been reported to be associated with the infection [4]. The current gold standard diagnosis of leptospirosis is the microagglutination test (MAT) [5]. The test usually requires live leptospires (approximately 20 serovars), expert personnel, and specific equipment, limiting its availability to only central or reference laboratories [6]. To overcome these challenges, several diagnostics such as enzyme-linked immunosorbent assays (ELISA),

polymerase chain reactions (PCR), and rapid lateral flow immunoassays (LFIs) have been developed [6,7].

In the last two decades, several LFIs have been created for the diagnosis of human leptospirosis and many of them have been commercialized [7]. The use of these LFIs in clinical settings, however, remains unpopular, and they are only applied as a screening test [7]. This is possibly due to the sensitivity and specificity of the tests which are inconsistent among studies, raising questions about the reliability and applicability of their use [7,8]. Thus, the primary objective of this study is to determine the overall sensitivity and specificity of the currently available human leptospirosis LFIs using systemic review and meta-analysis. Additionally, a secondary objective is set to investigate factors that may affect the assay accuracy or heterogeneity of test results. The investigated factors include brand, detection target, and phase of infection. We expected that our findings would be helpful for not only healthcare providers but also researchers or inventors developing LFIs for the diagnosis of human leptospirosis.

## Methods

This systematic review was conducted in accordance with the Preferred Reporting of Systematic Review Meta-analyses (PRISMA) guidelines [9]. The protocol was registered with the International Prospective of Systematic Reviews (PROSPERO registration number CRD42022371788).

### Search strategy

We searched the following databases for relevant literature: PubMed, Embase, and Scopus. The search was first conducted on December 1st, 2022 and latest updated on January 28th, 2024. Restrictions were applied exclusively to English literature, but not for publication date or country of study. A search strategy was constructed with three key concepts to identify studies that report the sensitivity and specificity of different lateral flow assays for leptospirosis diagnosis (**S1 Appendix**). An additional screening from the reference lists of included studies and published systematic reviews was also performed to identify relevant studies.

### Eligibility criteria

Original research articles either prospective or retrospective studies that met the following conditions were included in the meta-analysis: i) included patients intended to be diagnosed with leptospirosis; ii) examined the diagnostic accuracy of LFI; iii) used MAT or immunofluorescent assay (IFA) as a reference test; and iv) reported true positive (TP), false positive (FP), true negative (TN), and false negative (FN) values, allowing for calculation of sensitivity and specificity. We decided to include studies using the IFA as a reference test because the technique is based on the same principle as MAT and has been routinely used in some countries [10–12]. Studies were excluded if one or more of the following criteria were met: i) published in a language other than English; ii) evaluations of assays other than LFI; iii) evaluations performed in animals; and iv) insufficient data for the calculation of sensitivity and specificity.

### Study selection and data extraction

All retrieved articles were imported to Rayyan, and duplicated studies were removed [13]. Two authors (TN and LL) screened and selected the articles independently based on title and abstract, followed by a full-text review. Any disagreements in selection were solved through discussion among all three researchers (TN, LL, and SN).

After piloting and standardization, the developed Microsoft Access form (Microsoft Corp., WA, USA) was used to extract data from the eligible studies. Data extraction was performed in duplicate by two authors (TN and LL) independently, and the collected data were cross-verified. Additionally, the relevant data from each selected article was summarized in a descriptive format (S2 Appendix).

## Quality assessment

The quality assessment of the included articles was carried out using the criteria modified from Maze et. al. [7]. The criteria developed are in accordance with the revised Quality Assessment of Diagnostic Accuracy Studies criteria (QUADAS-2) [14]. For each selected study, the risk of bias was assessed in four domains: patient selection, index test, reference test, and flow and timing, using criteria listed in S1 Table. The study applicability was graded in the domains of patient selection, index test, and reference test, using the criteria listed in the S2 Table. The studies were scored as "low risk or no concern, +" if the criteria were met; "high risk or concern, -" if one of the criteria was not met; or "unclear, ?" if the information reported in the studies was not sufficient for assessment.

## Statistical analysis

**Estimates and plots.** The meta-analysis for this study was carried out using STATA version 16.1 (College Station, TX, USA) with all relevant packages (metandi, midas, and mylabels) installed. The extracted or calculated TP, FP, FN, and TN values from each study were included in the dataset. The primary measures are pooled sensitivity and specificity. We also plotted sensitivity and specificity data in forest plots calculated using a random effect bivariate model, and graphed study-specific estimates of sensitivity and specificity with 95% confidence intervals (CIs) in the summary receiver operating characteristic (SROC) curves, and the area under curve (AUC). To identify influential studies, the spike plot with Cook's distance was used. Outlier detection was performed by interpreting the standardized level 2 residuals for sensitivity and specificity data of the studies included in the quantitative analysis.

**Heterogeneity and subgroup analysis.** We used a bivariate box plot to evaluate the spread and skewness of the data. We also examined heterogeneity across studies by visually inspecting the forest plots. Subgroup meta-analysis was performed for the selected covariates: brand, detection target, and phases of infection.

**Publication bias.** We assessed publication bias for diagnostic tests using Deeks' funnel plot asymmetry test: a scatter plot of the inverse of the square root of the effective sample size [1/root (ESS)] *versus* the diagnostic log odds ratio (lnDOR). A symmetrical funnel shape with a *p-value* > 0.05 is considered to have no substantial publication bias.

## Results

A total of 459 records were identified from an electronic search. After removing duplicates, 323 records remained for screening titles and abstracts. Of these, 42 reports were considered for full-text evaluation. After the application of inclusion and exclusion criteria, 18 reports were excluded, giving a total of 24 reports eligible for the review and meta-analysis (Fig 1) [11,15–37]. The records that were excluded and reasons for exclusion are listed in S3 Table [38–55].

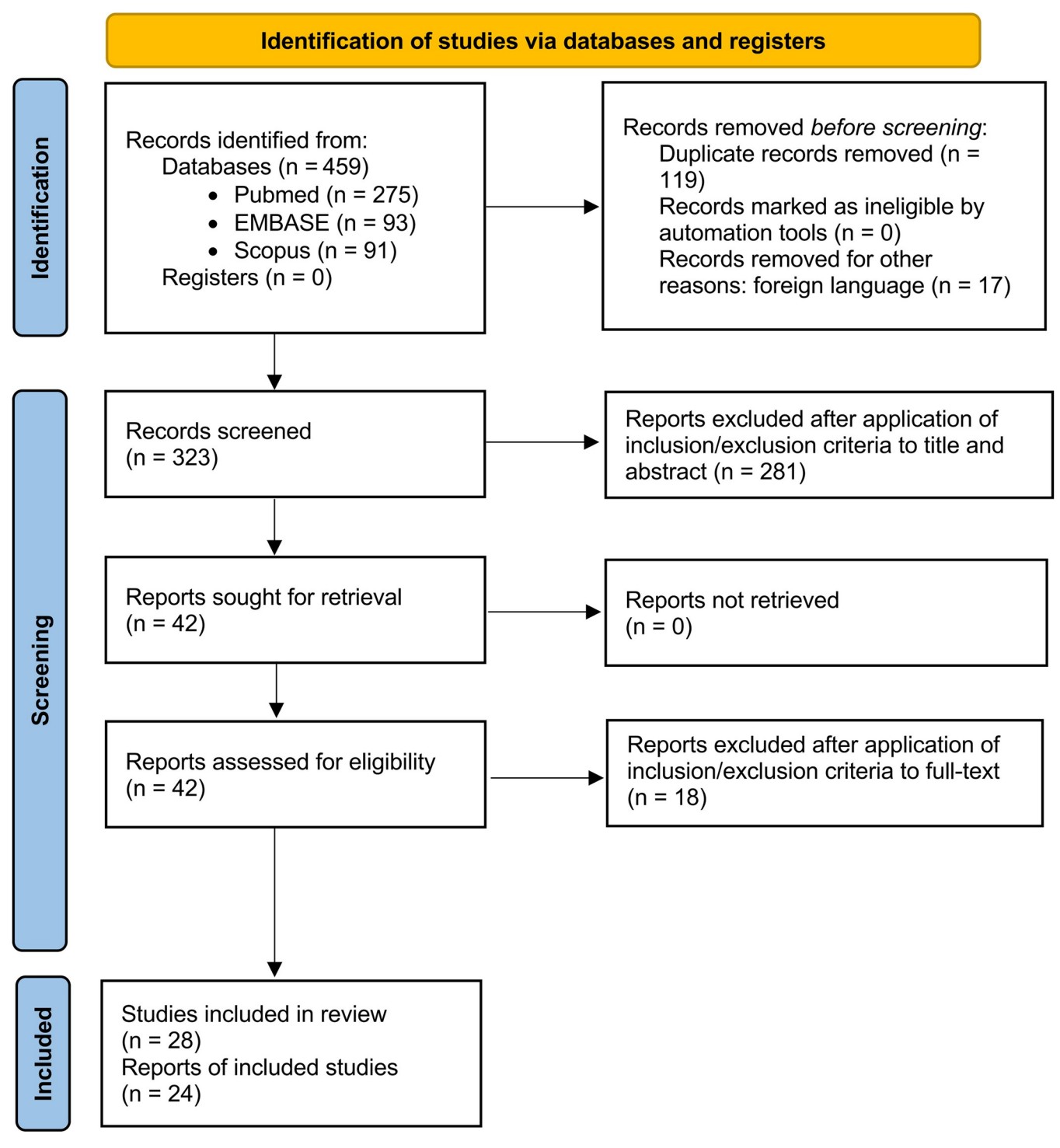

**Fig 1. The PRISMA flow diagram.**

## Characteristics of eligible studies

The articles or reports included in this review are listed in **Table 1**. These reports were published from 2003 through 2023. There are 28 studies extracted from 24 reports. This is because

**Table 1. List of the articles included in this review.**

| First author, published year | Country | Study design | Sample size | Reference test | LFI test | Ref. |
|---|---|---|---|---|---|---|
| Jirawannaporn, 2023 | Thailand | Prospective and retrospective | 171 | MAT, blood culture, and RT-PCR | Medical Science Public Health | [36] |
| Campos, 2023 | Brazil | Unclear | 200 | MAT | IN-LFI | [37] |
| Bottieau, 2022 | Sudan, Congo, Nepal, and Cambodia | Prospective | 1922 | MAT and PCR | Test-IT Leptospira IgM | [15] |
| | | | | | SD Bioline Leptospira IgG/IgM | |
| Dinhuzen, 2021 | Thailand | Prospective | 99 | MAT, blood culture, and RT-PCR | Medical Science Public Health | [16] |
| | | | | | Leptocheck WB | |
| | | | | | SD Bioline leptospirosis | |
| | | | | | TRUSTline | |
| | | | | | J.Mitra | |
| Silpasakorn, 2020 | Thailand | Retrospective | 434 | IFA | ImmuneMed AFI rapid | [11] |
| Dawson, 2020 | Federated States of Micronesia | Prospective | 91 | MAT | Bioline Leptospira IgM | [17] |
| Rao, 2019 | Malaysia | Retrospective | 142 | MAT | Leptocheck WB | [19] |
| Alia, 2019 | Malaysia | Prospective | 50 | MAT and qPCR | Leptocheck WB | [18] |
| | | | | | ImmuneMed Leptospira IgM Duo Rapid Test | |
| | | Retrospective | 135 | MAT | Leptocheck WB | |
| | | | | | ImmuneMed Leptospira IgM Duo Rapid Test | |
| Amran, 2018 | Malaysia | Retrospective | 197 | MAT | ImmuneMed Leptospira IgM Duo Rapid Test | [20] |
| Nabity, 2018 | Brazil | Prospective | 74 | MAT and culture | Dual Path Platform | [21] |
| Doungchawee, 2017 | Thailand | Retrospective | 168 | MAT and culture | LEPkit | [22] |
| Lee, 2016 | Korea | Retrospective | 127 | MAT | ImmuneMed Leptospira Rapid Test | [23] |
| | Bulgaria | Retrospective | 50 | MAT | | |
| | Argentina | Retrospective | 208 | MAT | | |
| Eugene, 2015 | Sri Lanka | Prospective | 84 | MAT | Leptocheck WB | [24] |
| Podgoršek, 2015 | Slovenia | Prospective | 590 | MAT, PCR, and culture | Leptocheck WB | [26] |
| Niloofa, 2015 | Sri Lanka | Prospective | 888 | MAT | Leptocheck WB | [25] |
| Colt, 2014 | Federated States of Micronesia | Prospective | 83 | MAT | SD Bioline Leptospira IgG/IgM | [28] |
| Chang, 2014 | Malaysia | Retrospective Retrospective | 128 | MAT, PCR | VISITECT Lepto | [27] |
| | | | 125 | MAT | | |
| Widiyanti, 2013 | Philippines | Prospective | 58 | MAT, PCR | ICG-based LFA | [30] |
| | | | | | dipsticks | |
| Goris, 2013 | The Netherlands | Prospective | 1404 | MAT, ELISA, and culture | LeptoTex Lateral Flow | [29] |
| | | | 2757 | | Leptocheck WB | |
| Nabity, 2012 | Brazil | Retrospective | 987 | MAT | Dual Path Platform | [31] |
| Silpasakorn, 2011 | Thailand | Retrospective | 161 | MAT and culture | SD Bioline leptospirosis | [32] |
| Cohen, 2007 | Thailand | Prospective | 706 | MAT | Multi-Test Dip_S-Ticks | [33] |
| Blacksell, 2006 | Laos | Prospective | 168 | MAT | Leptotek | [34] |
| Sehgal, 2003 | India | Prospective | 117 | MAT and culture | Lepto Lateral flow | [35] |

the reports published by Alia et. al., Lee et. al., and Chang et. al. presented the results from two, three, and two studies, respectively [18,23,27]. Studies were conducted among patients from Argentina, Brazil, Bulgaria, Cambodia, Congo, India, Korea, Laos, Malaysia, Micronesia, Nepal, Netherlands, Philippines, Slovenia, Sri Lanka, Sudan, and Thailand. Among all 28

studies, 12 studies (42.8%) are retrospective studies in which the evaluations were performed using archived samples. The number of patients or samples varied from 50 to approximately 2,800. All the studies used MAT (solely or along with other tests) as a reference test, except one study published by Silpasakorn et. al., in which only IFA was used to define leptospirosis cases [11]. In addition, variety in case definition was observed among the studies (S4 Table). In all 24 reports eligible for this meta-analysis, 16 different commercial and 4 different in-house LFIs were evaluated. However, it should be noted that, in this study, the articles published in non-English languages were excluded. Thus, the total number of 20 LFIs included in this review may not correspond to the number of all developed LFIs.

## Study quality

The results of the quality assessment were summarized in Table 2. The assessment was carried out using the criteria presented in the S1 and S2 Tables. The results showed that only nine studies (32.1%) were considered "low risk" of bias and "no concern" for applicability. Thirteen studies (46.4%) were considered "low risk" of bias for all four domains. Eleven studies (39.3%)

Table 2. Quality assessment of the studies included in this review.

| First author, published year | Risk of bias | | | | Applicability | | | Ref. |
|---|---|---|---|---|---|---|---|---|
| | Patient selection | Index test | Reference test | Flow and timing | Patient selection | Index test | Reference test | |
| Jirawannaporn, 2023 | + | + | + | + | + | + | + | [36] |
| Campos, 2023 | + | + | ? | ? | + | + | ? | [37] |
| Bottieau, 2022 | + | + | + | + | + | + | ? | [15] |
| Dinhuzen, 2021 | + | + | + | + | + | + | + | [16] |
| Silpasakorn, 2020 | - | + | + | + | + | + | - | [11] |
| Dawson, 2020 | + | + | + | + | + | + | ? | [17] |
| Rao, 2019 | + | + | + | + | + | + | + | [19] |
| Alia, 2019 (prospective) | + | + | + | - | + | + | + | [18] |
| Alia, 2019 (retrospective) | - | + | - | + | + | + | + | [18] |
| Amran, 2018 | + | + | + | + | + | + | + | [20] |
| Nabity, 2018 | + | + | + | + | + | + | + | [21] |
| Doungchawee, 2017 | - | + | + | ? | + | + | + | [22] |
| Lee, 2016 (Korea) | - | + | + | ? | + | + | + | [23] |
| Lee, 2016 (Bulagaria) | - | + | + | ? | + | + | + | [23] |
| Lee, 2016 (Argentia) | - | + | + | ? | + | + | + | [23] |
| Eugene, 2015 | + | + | - | + | + | + | - | [24] |
| Podgoršek, 2015 | + | + | + | + | + | + | + | [26] |
| Niloofa, 2015 | + | + | + | + | + | + | - | [25] |
| Colt, 2014 | + | + | + | + | + | + | + | [28] |
| Chang, 2014 (acute) | - | + | - | ? | + | + | + | [27] |
| Chang, 2014 (mixed) | - | + | + | ? | + | + | + | [27] |
| Widiyanti, 2013 | - | + | + | + | + | + | + | [30] |
| Goris, 2013 | + | + | + | + | + | + | + | [29] |
| Nabity, 2012 | - | + | + | + | + | + | + | [31] |
| Silpasakorn, 2011 | - | + | + | + | + | + | - | [32] |
| Cohen, 2007 | + | + | + | + | + | + | ? | [33] |
| Blacksell, 2006 | + | + | + | + | + | + | + | [34] |
| Sehgal, 2003 | + | + | + | + | + | + | + | [35] |

"+" = low risk or no concern; "-" = high risk or concern; "?" = unclear

were classified as "high risk" of bias for patient selection because healthy participants were used as controls; ten of them are retrospective case-control selection studies. Three studies (10.7%) were rated "high risk" of bias for reference tests since MAT was performed only on acute-phase sera. Seven studies (25.0%) were graded "unclear" for flow and timing domain because it is unclear whether samples in each study were subjected to the same reference tests. In terms of applicability, all 28 studies were classified as "no concern" on patient selection and index test domains. However, four studies (14.3%) were rated "unclear" and four studies (14.3%) were rated "concern" in the reference test domain. This is because the serovar panel used in MAT test in those studies was not listed or did not cover serovars circulating in the areas, respectively.

## Descriptive analysis

All LFIs that were evaluated in this review are listed in the **S5 Table** and the evaluation results are summarized in **Table 3**. Among the 20 LFIs evaluated, 18 were designed to detect *Leptospira*-specific antibodies in blood or serum samples, whereas the other two assays are antigen detection targeting leptospiral lipopolysaccharide (LPS) in patient urines. These antigen detection assays have a sensitivity of 80–89% and a specificity of 74–87% [30]. Most antibody detection LFIs (11/18) aim to detect specific IgM. All these IgM detection assays were designed to have one reading window with a single control line and a single test line, except the ImmuneMed Leptospira IgM Duo Rapid Test that was developed to contain two reading windows [20]. This design allows samples to be tested at two different cutoff titers (1:50 and 1:200). The second most common design (5/18) for the antibody detection test strips is IgM/IgG detection. Of these, only the ImmuneMed Leptospira Rapid Test, was designed to contain two separate reading windows for IgM and IgG, while the remaining were constructed with three lines (IgM, IgG, and control lines) on the same strip [23]. The variety of designs of leptospirosis LFIs also includes ImmuneMed AFI rapid Test that was developed in a multiplex format to detect leptospirosis simultaneously with scrub typhus and hemorrhagic fever [11]. Additionally, the Dual Path Platform from Chembio Diagnostic Systems, USA, was designed in such a way that the sample and chase buffer were flown perpendicularly [21,31]. The accuracy of these LFIs were summarized in **Tables 3** and **S6**.

In addition to the variety of LFI configurations described above, we also looked at the leptospiral antigens that were selected to be incorporated into the LFIs. Among the 18 antibody detection LFIs evaluated in this review, only 11 LFIs have disclosed the antigen they used. *L. interrogans* heat extract was found to be the most common antigens employed in the LFIs, followed by polysaccharide from nonpathogenic *L. biflexa* serovar Patoc. Of these, recombinant protein has been used in only one LFI [21,31].

## Meta-analysis

Of these 28 studies extracted from 24 reports, several studies evaluated LFIs using samples collected from different phases of infection i.e., acute phase sera (mainly collected within 10 days after onset) and convalescent phase sera (mainly collected within two weeks after acute phase). In addition, the study conducted by Nabity et. al. evaluated the accuracy of the Dual Path Platform from Chembio Diagnostic Systems using samples collected by three different means i.e., finger stick blood (FSB), vein whole blood (VWB), and serum [21]. Thus, multiple sensitivity and specificity values were reported from these studies. As a result, a total of 49 data entries were employed for meta-analysis (**Table 3**).

**Pooled sensitivity and specificity.** We found that the sensitivity of individual entries varied widely, from 3.6% to 100.0%. The range of specificity for each entry was narrower, ranging

**Table 3. Summary of diagnostic accuracy of LFIs evaluated in this review.**

| Entry ID | First author, published year | Lateral Flow Immunoassay | Detection | Phase | Sens. | Spec. | TP | FP | TN | FN | Remark | Ref. |
|---|---|---|---|---|---|---|---|---|---|---|---|---|
| 1 | Dawson, 2020 | Bioline Leptospira IgM | IgM | no data | 100 | 67 | 5 | 28 | 58 | 0 | | [17] |
| 2 | Dinhuzen, 2021 | TRUSTline | IgM/IgG | acute | 33.9 | 88.4 | 19 | 5 | 38 | 37 | | [16] |
| 3 | Nabity, 2018 | Dual Path Platform | antibody* | acute | 92.6 | 80.4 | 25 | 9 | 37 | 2 | FSB# | [21] |
| 4 | Nabity, 2018 | | antibody | acute | 96.3 | 74.5 | 26 | 12 | 35 | 1 | VWB$ | [21] |
| 5 | Nabity, 2018 | | antibody | acute | 85.2 | 87.2 | 23 | 6 | 41 | 4 | serum | [21] |
| 6 | Nabity, 2012 | | antibody | acute | 82.6 | 93.8 | 256 | 42 | 635 | 54 | | [31] |
| 7 | Nabity, 2012 | | antibody | convalescent | 89 | 93.8 | 121 | 42 | 635 | 15 | | [31] |
| 8 | Dinhuzen, 2021 | Medical Science Public Health | IgM | acute | 60.7 | 65.1 | 34 | 15 | 28 | 22 | | [16] |
| 9 | Jirawannaporn, 2023 | | IgM | mixed | 55.67 | 63.08 | 59 | 24 | 41 | 47 | | [36] |
| 10 | Silpasakorn, 2020 | ImmuneMed AFI Rapid | IgM | acute | 37.4 | 99 | 49 | 3 | 300 | 82 | | [11] |
| 11 | Silpasakorn, 2020 | | IgG | acute | 9.2 | 100 | 12 | 0 | 303 | 119 | | [11] |
| 12 | Silpasakorn, 2020 | | IgM/IgG | acute | 38.2 | 99 | 50 | 3 | 300 | 81 | | [11] |
| 13 | Silpasakorn, 2020 | | IgM | convalescent | 84.6 | 96.2 | 77 | 4 | 100 | 14 | | [11] |
| 14 | Silpasakorn, 2020 | | IgG | convalescent | 47.3 | 100 | 43 | 0 | 104 | 48 | | [11] |
| 15 | Silpasakorn, 2020 | | IgM/IgG | convalescent | 84.6 | 96.2 | 77 | 4 | 100 | 14 | | [11] |
| 16 | Alia, 2019 | ImmuneMed Leptospira IgM Duo Rapid Test | IgM | acute | 15.8 | 90.3 | 3 | 3 | 28 | 16 | prospective | [18] |
| 17 | Alia, 2019 | | IgM | acute | 40.2 | 89.5 | 39 | 4 | 34 | 58 | retrospective | [18] |
| 18 | Amran, 2018 | | IgM | no data | 73 | 90 | 68 | 10 | 94 | 25 | | [20] |
| 19 | Lee, 2016 | ImmuneMed Leptospira Rapid Test | IgM/IgG | acute | 93.9 | 97.9 | 31 | 2 | 92 | 2 | Korea | [23] |
| 20 | Lee, 2016 | | IgM/IgG | acute | 100 | 100 | 25 | 0 | 25 | 0 | Bulgaria | [23] |
| 21 | Lee, 2016 | | IgM/IgG | acute | 81 | 95.4 | 81 | 5 | 103 | 19 | Argentina | [23] |
| 22 | Dinhuzen, 2021 | J.Mitra | IgM/IgG | acute | 3.6 | 97.7 | 2 | 1 | 42 | 54 | | [16] |
| 23 | Bottieau, 2022 | Test-IT Leptospira IgM | IgM | convalescent | 16 | 95.8 | 5 | 34 | 785 | 26 | | [15] |
| 24 | Chang, 2014 | VISITECT Lepto | IgM | acute | 24 | 94 | 14 | 4 | 66 | 44 | | [27] |
| 25 | Chang, 2014 | | IgM | mixed | 40 | 94 | 22 | 4 | 66 | 33 | | [27] |
| 26 | Goris, 2013 | LeptoTek Lateral Flow | IgM | mixed | 78 | 95 | 85 | 66 | 1229 | 24 | | [29] |
| 27 | Blacksell, 2006 | | IgM | convalescent | 47.3 | 75.5 | 11 | 40 | 123 | 12 | | [34] |
| 28 | Blacksell, 2006 | | IgM | acute | 70 | 75 | 7 | 15 | 45 | 3 | | [34] |
| 29 | Goris, 2013 | | IgM | acute | 69 | 96 | 74 | 57 | 1235 | 34 | | [29] |
| 30 | Cohen, 2007 | Multi-Test Dip_S-Ticks, DSLST | IgM | convalescent | 82 | 81 | 55 | 121 | 516 | 82 | | [33] |
| 31 | Bottieau, 2022 | SD Bioline Leptospira IgG/IgM | IgM/IgG | convalescent | 9 | 97.4 | 6 | 49 | 1804 | 58 | | [15] |
| 32 | Colt, 2014 | | IgM/IgG | mixed | 69.2 | 90 | 9 | 7 | 63 | 4 | | [28] |
| 33 | Dinhuzen, 2021 | SD Bioline Leptospirosis | IgG | acute | 1.8 | 93 | 1 | 3 | 40 | 55 | | [16] |
| 34 | Silpasakorn, 2011 | | IgG | mixed | 83.2 | 98.6 | 74 | 1 | 71 | 15 | | [32] |
| 35 | Sehgal, 2003 | Lepto Lateral Flow | IgM | acute | 52.9 | 93.6 | 37 | 3 | 44 | 33 | | [35] |
| 36 | Sehgal, 2003 | | IgM | convalescent | 86 | 89.4 | 49 | 5 | 42 | 8 | | [35] |
| 37 | Dinhuzen, 2021 | Leptocheck WB | IgM | acute | 75 | 53.5 | 42 | 20 | 23 | 14 | | [16] |
| 38 | Rao, 2019 | | IgM | no data | 66.6 | 78.9 | 44 | 16 | 60 | 22 | | [19] |
| 39 | Alia, 2019 | | IgM | acute | 47.4 | 80.7 | 9 | 6 | 25 | 10 | prospective | [18] |
| 40 | Alia, 2019 | | IgM | acute | 90.7 | 76.3 | 88 | 9 | 29 | 9 | retrospective | [18] |
| 41 | Eugene, 2015 | | IgM | acute | 95 | 76.4 | 34 | 12 | 32 | 6 | | [24] |
| 42 | Podgoršek, 2015 | | IgM | acute | 80 | 98.6 | 28 | 8 | 547 | 7 | | [26] |
| 43 | Niloofa, 2015 | | IgM | acute | 80.8 | 76.9 | 286 | 121 | 405 | 76 | | [25] |
| 44 | Goris, 2013 | | IgM | mixed | 78 | 98 | 153 | 63 | 2497 | 44 | | [29] |
| 45 | Goris, 2013 | | IgM | acute | 55 | 98 | 100 | 56 | 2495 | 83 | | [29] |
| 46 | Widiyanti, 2013 | Dipsticks | LPS | mixed | 80 | 74 | 28 | 6 | 17 | 7 | urine | [30] |

*(Continued)*

**Table 3.** (Continued)

| Entry ID | First author, published year | Lateral Flow Immunoassay | Detection | Phase | Sens. | Spec. | TP | FP | TN | FN | Remark | Ref. |
|---|---|---|---|---|---|---|---|---|---|---|---|---|
| 47 | Widiyanti, 2013 | ICG-based LFA | LPS | mixed | **89** | **87** | 31 | 3 | 20 | 4 | urine | [30] |
| 48 | Doungchawee, 2017 | LEPkit | IgM | acute | **97.4** | **94.5** | 75 | 5 | 86 | 2 | | [22] |
| 49 | Campos, 2023 | IN-LFI | IgM | acute | **97** | **97** | 97 | 3 | 97 | 3 | | [37] |

Sens. = sensitivity; Spec. = specificity; TP = true positive; FP = false positive; TN = true negative; FN = false negative

* It is unknown if the test was designed to detect IgM or IgG.

# FSB = finger stick blood, indicating that the results were obtained from finger stick blood samples

$ VWB = venous whole blood, indicating that the results were obtained from venous whole blood samples

from 53.5% to 100.0%. Firstly, the estimation of the pooled sensitivity and specificity was calculated by a random effect bivariate model with all 49 data entries included. The entry-specific results were displayed in the forest plot (**Fig 2**). The estimated pooled sensitivity and specificity were 68% (95% confidence interval, CI: 57–78) and 93% (95% CI: 90–95), respectively. The $I^2$ values for sensitivity and specificity were 95.76% and 96.35%, respectively, indicating that the data entries included might be too heterogeneous to achieve accurate meta-analysis results. The AUC from the SROC curve was 0.92 (**Fig 3**). The Deeks' funnel plot asymmetry showed no potential publication bias (**Fig 4**). The residual-based goodness-of-fit and bivariate

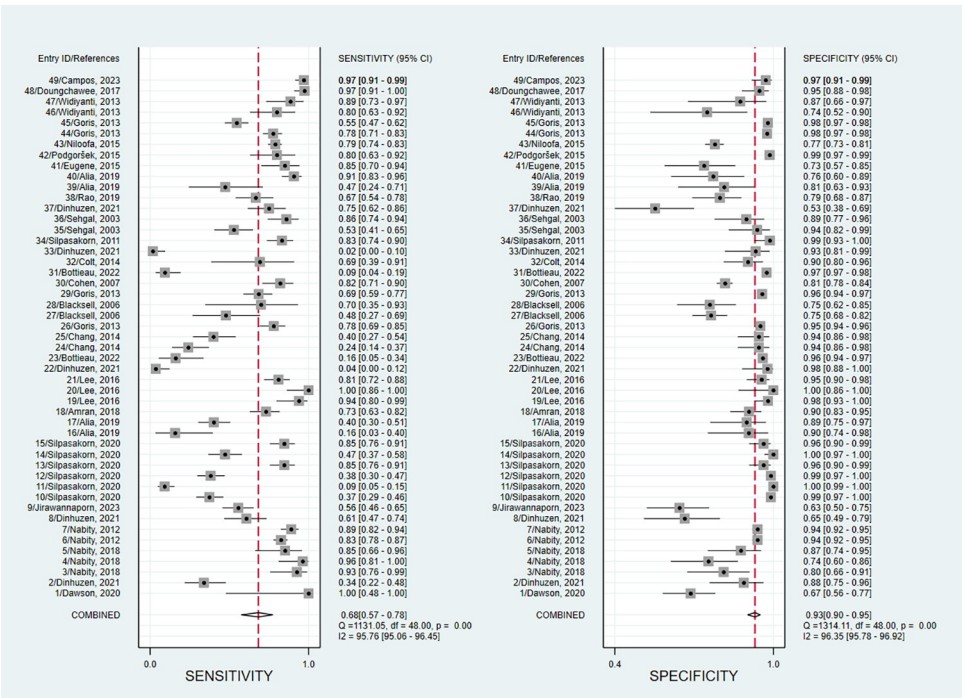

**Fig 2. Forest plot showing entry-specific and mean sensitivity and specificity with corresponding heterogeneity statistics.** Forest plots of the analysis about the prediction value of LFI kits for leptospirosis in terms of sensitivity and specificity with the data of all 49 entries. Square symbols represent the sensitivity or specificity of each study according to the entry ID shown on the y-axis, while the short lines cutting through represent the relative 95% CI. The diamond symbols refer to the combined sensitivity or specificity, which was automatically calculated and displayed by Stata software. A "COMBINED" label corresponding to the diamond symbol is shown on the y-axis underneath all entry IDs.

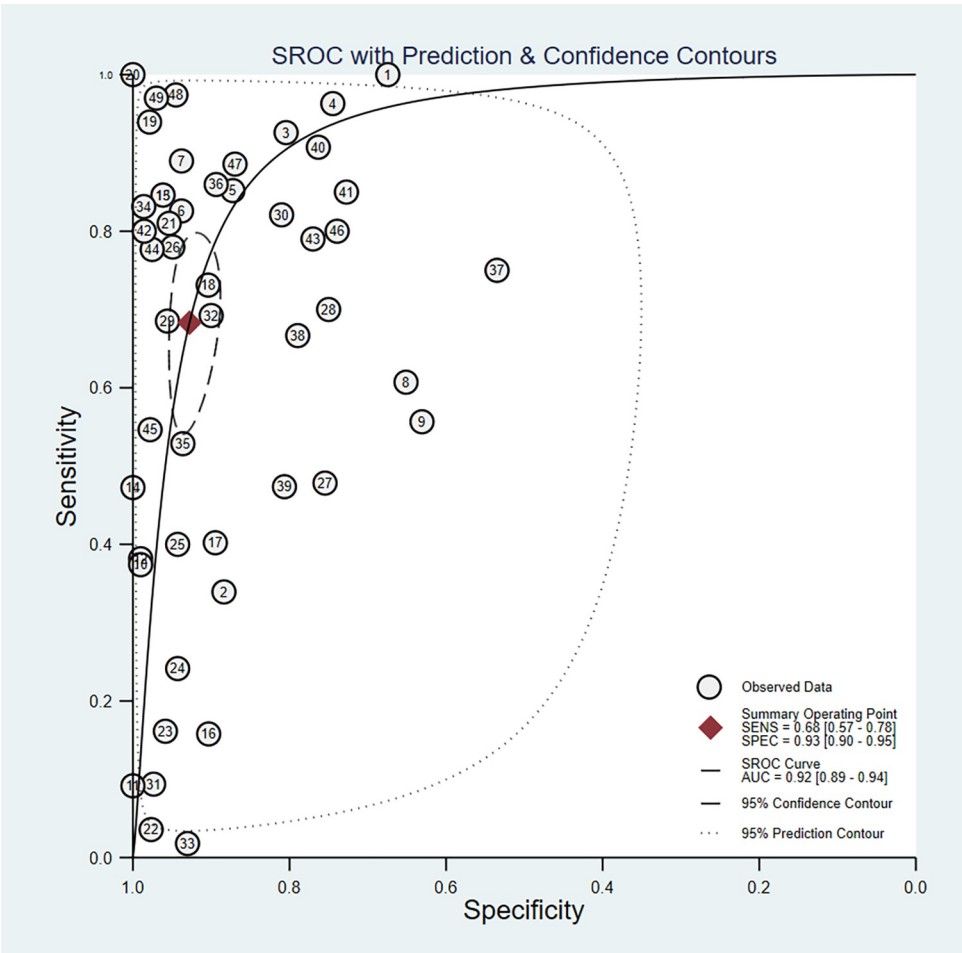

**Fig 3. Summary ROC curve with confidence and prediction regions around mean operating sensitivity and specificity points.**

normality plots suggested that the random effect model used was applicable. The influence and outlier detection analyses, however, identified 3 common outliers (**S1 Fig**). Next, we performed sensitivity and specificity analysis by removing the outlier that was graded high risk during the study quality assessment (entry ID 11) and found that the pooled sensitivity and specificity did not change significantly (**S2 Fig**). Thirdly, we calculated the combined sensitivity and specificity values using only one data entry per study. In this case, sensitivity and specificity obtained from acute phase samples were chosen as representative for the studies reporting multiple data entries. The combined sensitivity and specificity were 66% (95% CI: 51–78) and 92% (95% CI: 88–94), respectively, which were comparable to those obtained from all data entries (**S3 Fig**).

**Subgroup analysis by LFI brands.** The sensitivity and specificity of each brand of LFI were calculated using subgroup analysis and summarized in the **S6 Table**. This analysis, however, was limited to LFIs that have four or more data entries available. The results showed that the Dual Path Platform had the highest sensitivity (90%; 95% CI: 82–94), followed by Lepto-check WB (75%; 95% CI: 66–82), and LeptoTek Lateral Flow (65%; 95% CI: 52–76). The highest specificity was of ImmuneMed AFI rapid (99%; 95% CI: 97–100), followed by Dual Path

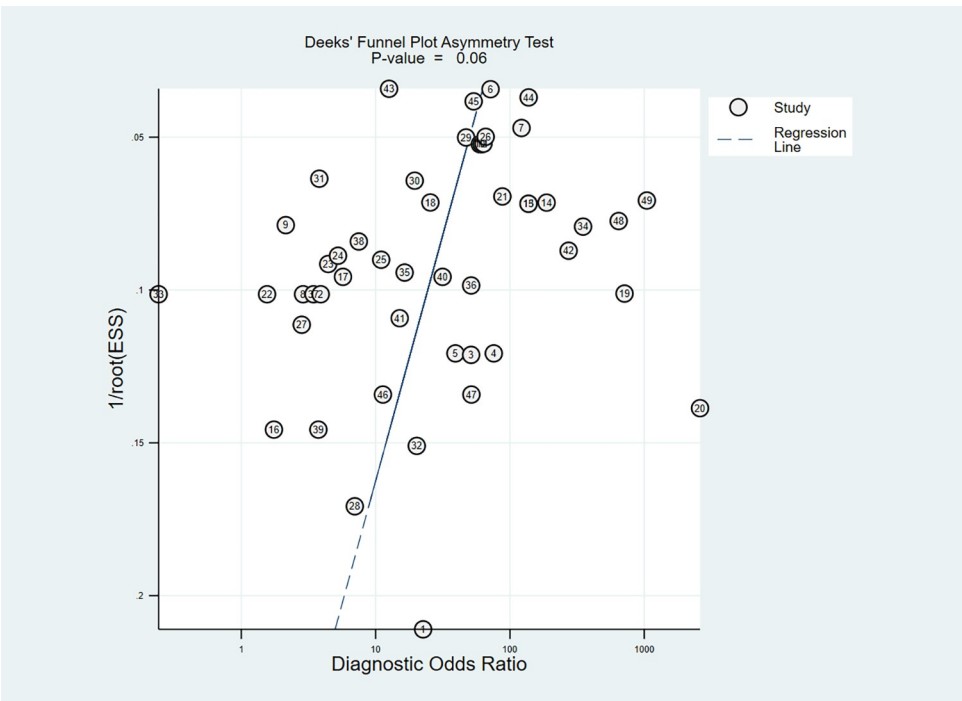

**Fig 4. Funnel plot with superimposed regression line (publication bias).** STATA command: midas tp fp fn tn, pubbias.

Platform (89%; 95% CI: 81–94), and LeptoTek Lateral Flow (89%; 95% CI: 76–95), respectively.

**Subgroup analysis by detection targets.**   We also performed subgroup analysis to investigate the influence of the detection targets (**S7 Table**). The LFIs that provided the greatest sensitivity are the IgM detection type (69%; 95% CI: 58–78), followed by the IgM/IgG detection type (62%; 95% CI: 26–88). IgG detection LFIs demonstrated the lowest sensitivity of 23% (95% CI: 3–73). On the contrary, the greatest specificity is of IgG detection LFIs (100%; 95% CI: 94–100), followed by IgM/IgG (97%; 95% CI: 94–98), and IgM detection (90%; 95% CI: 86–94), respectively.

**Subgroup analysis by infection phases.**   The pooled sensitivity and specificity as grouped by infection phases were also determined (**S8 Table**). When samples collected from acute phase, convalescent phase, and mixed samples were used to evaluate the LFIs, the pooled sensitivities of the LFIs were 67% (95% CI: 50–80), 64% (95% CI: 39–83), and 69% (95% CI: 56–80), and the pooled specificities were 93% (95% CI: 88–96), 94% (95% CI: 89–97), and 94% (95% CI: 85–97), respectively.

## Discussion

In this study, original articles evaluating the human leptospirosis diagnostic LFIs were systematically recruited for review and meta-analysis. The recruited articles were restricted to those that used MAT as a reference test because MAT is considered a gold standard diagnosis for leptospirosis and has been used globally [3,6]. We also included studies with IFA as a reference test because the technique is based on the same antigen-antibody reaction as MAT and has been used routinely in some endemic areas [10–12].

Meta-analysis was initially performed with all 49 data entries (**Table 3**) extracted from 28 eligible studies (**Table 1**). We found that the estimated pooled sensitivity and specificity were 68% and 93%, respectively (**Fig 2**). However, we thought that samples from the acute phase of infection were more relevant to real-world settings, as patients tend to visit hospitals a few days after onset. The pooled sensitivity and specificity were recalculated regarding this matter, and the results are not significantly different compared to the initial analysis. We noted that the pooled specificity of LFIs is somewhat acceptable, but the pooled sensitivity is low [56]. Nevertheless, it is important to emphasize that the estimated pooled sensitivity and specificity should be interpreted with caution due to the substantial heterogeneity observed among the included studies, despite the adoption of a random effect model. Furthermore, the forest plot's $I^2$ values indicate that the included studies were considerably heterogeneous to produce dependable meta-analysis results. Several potential factors contributing to heterogeneity were spotted during data extraction.

**First**, we identified that the study designs could be a potential cause of the data heterogeneity. Two types of study designs were observed among the eligible studies: a case-control type and a cohort type cross-sectional study [57]. We found that 17 studies (61%) were cohort type cross-sectional studies. These were considered low risk for patient selection bias. The others 11 studies (39%), instead, were case-control type and were graded as high risk of bias in patient selections.

**Second**, reference tests and case definitions varied dramatically across the studies (**S4 Table**). For examples, some studies used MAT or IFA solely as a reference (54%) while others (46%) used MAT together with other methods. Also, the *Leptospira* serovar panel used in MAT might not cover all the predominant serovars in some studies. While several studies defined leptospirosis cases based on a four-fold increase in MAT antibody titers together with single MAT antibody titers, some evaluations used either single titers or paired titers only to define the cases. We noted that the investigators might define the single MAT titer cutoffs differently based on the prevalence of leptospirosis in the areas, and these cutoffs could be distinct from the WHO recommended cutoff of 1:400 [7,58]. However, various single MAT titer cutoffs within the same settings were detected (**S4 Table**). We determined that 10 studies (36%) were at risk (high risk and unclear) of bias and applicability for the reference test domain.

**Third**, we found that the samples themselves could also cause the heterogeneity of the evaluation results. While some investigators used samples collected from the acute phase of infection to evaluate the LFIs, others might use convalescent or mixed samples. Principally, acute phase samples contain a lower level of *Leptospira*-specific antibodies, and they are mainly IgM. Convalescent phase samples, on the contrary, have a higher titer of *Leptospira*-specific IgG [6]. Thus, even the same assays could have different sensitivity and specificity if different phases of samples were used for the evaluation.

**Fourth**, we also found that detection targets varied among the investigated LFIs. These variations include LPS detection LFIs, IgM detection LFIs, IgG detection LFIs, and combined IgM/IgG detection LFIs. Since the presence of IgM and IgG in the samples is determined by the phase of infection, evaluations of IgG detection LFIs with acute phase samples, for example, could give an expected lower sensitivity as compared to IgM detection LFIs [6]. This demonstrates that the detection target of LFI, among other factors, impacts the test performance.

**Fifth**, in addition to detection target, differences between each brand of the investigated LFIs also include: i) test designs (single-plex, multiplex, or two reading-windows formats); ii) flow patterns (regular lateral flow or perpendicular flow); iii) types of antigens incorporated (heat extract of the bacteria, purified LPS, recombinant proteins, etc.); and iv) miscellaneous factors such as types of membranes, amounts of antigen or antibody on the test and control lines, particle size of conjugated gold, etc. As a result, different brands or models of LFIs could perform differently due to the variations in these elements.

A question that is often asked in clinical settings (especially when there are several tests available, but test performances are heterogeneous among studies) is which brand of LFIs has the highest accuracy. To answer this question, we investigated further using multivariate meta-regression and subgroup analysis. However, this analysis is limited to the Dual Path Platform, Leptocheck WB, LeptoTek Lateral Flow, and ImmuneMed AFI Rapid only due to the availability of the data. The analysis results illustrated that the sensitivity and specificity are statistically comparable between each brand of LFIs (S6 Table).

Additionally, we conducted another subgroup analysis to understand how the LFIs performed with samples from different phases of infection. The results demonstrated that the sensitivity and specificity are practically unchanged, suggesting that the LFIs could be used with samples collected from either phase (S8 Table). Initially, this finding surprised us because acute phase samples are considerably not suitable for the IgG detection LFIs in principle. This is because the specific IgG has not been raised yet during the acute phase of the infection. However, when we examined the data closely, we found that the majority of the investigated LFIs are IgM and IgM/IgG detection types. Therefore, in this analysis, the results from IgM-targeted LFIs probably masked those from IgG detection types and gave the overall results as reported. We also found it interesting that the sensitivity of IgM detection LFIs remained unaffected when convalescent samples were used. Theoretically, the level of specific IgM declines during the convalescent phase of infections, causing IgM-targeted diagnostics to become less sensitive with convalescent samples. However, it has been reported that the specific IgM may stay in leptospirosis patients for months, which means that IgM-targeted LFIs will react with the convalescent samples [59].

In this study, subgroup analysis on detection targets was also carried out. As expected, IgM detection LFIs demonstrated a higher sensitivity over IgG detection types whereas specificity seems to be unaffected by this covariate (S7 Table). This finding would suggest that the IgM detection type should be chosen for further developments of the leptospirosis diagnostic LFIs.

In summary, during the past two decades, at least 20 LFIs for the diagnosis of human leptospirosis have been developed and marketed. Several studies were conducted to evaluate those LFIs; however, the results were markedly varied. In this study, we attempted to gain a better understanding of the accuracy of the available LFIs using a systematic review and meta-analysis. We found that the available data are heterogeneous; thus, the estimated accuracy of the LFIs derived from this study may not be at most reliable, especially the pooled sensitivity of the assays. This finding is in agreement with the systematic review and meta-analysis studies published previously [7,8]. The sources of heterogeneity in the data potentially came from both LFIs themselves and evaluation procedures. Variations in evaluation procedures seem to contribute to both assay sensitivity and accuracy. On the basis of our findings, it is difficult to ascertain whether the LFIs are clinically useful. Sequentially, we recommend that further evaluations of LFIs are needed, as also suggested by the previous reviews [7,8]. Further evaluations should be conducted as per standard guidelines, such as the Standard for Reporting Diagnostic Accuracy Studies (STARD) [60]. Additionally, they should use reference tests with sensitivity and specificity close to 100% or statistical tools that account for imperfect reference test accuracy, e.g., Bayesian latent class model [10]. Diversity in LFI designs, on the other hand, is likely to primarily affect the sensitivity of the assays. Our finding suggests that IgM or combined IgM/IgG detection types of the LFIs would be more suitable for the diagnosis of leptospirosis as compared to IgG detection types. However, in this current study, the comparison between antigen detection and antibody detection LFIs could not be done because the data from the antigen detection type is not sufficient.

## Supporting information

**S1 Checklist. PRISMA-DTA.**
(DOCX)

**S2 Checklist. PRISMA DTA for abstract.**
(DOCX)

**S1 Table. Criteria for assessing risk of bias of studies included in this review.**
(DOCX)

**S2 Table. Criteria for assessing the applicability of the studies included in this review.**
(DOCX)

**S3 Table. Reasons for exclusion.**
(DOCX)

**S4 Table. List of reference tests and case definitions used in the selected studies.**
(DOCX)

**S5 Table. List of LFI evaluated by the studies included in this review.**
(DOCX)

**S6 Table. Subgroup analysis by brand of LFI.**
(DOCX)

**S7 Table. Subgroup analysis by detection targets.**
(DOCX)

**S8 Table. Subgroup analysis by phases of infection.**
(DOCX)

**S1 Fig.** Graphical depiction of influence (A) and outlier detection (B) analyses.
(TIFF)

**S2 Fig. Forest plot showing mean sensitivity, specificity, and corresponding heterogeneity statistics with all data entries except the high-risk outlier (entry ID 11).**
(TIFF)

**S3 Fig. Forest plot showing mean sensitivity, specificity, and corresponding heterogeneity statistics with data limited to one data entry per study.**
(TIFF)

**S1 Appendix. Search strategy.**
(DOCX)

**S2 Appendix. Summary of individual reports included in the review.**
(DOCX)

## Author Contributions

**Conceptualization:** Teerapat Nualnoi, Supawadee Naorungroj.

**Data curation:** Teerapat Nualnoi, Supawadee Naorungroj.

**Formal analysis:** Supawadee Naorungroj.

**Funding acquisition:** Teerapat Nualnoi.

**Investigation:** Teerapat Nualnoi, Luelak Lomlim, Supawadee Naorungroj.

**Methodology:** Teerapat Nualnoi, Luelak Lomlim, Supawadee Naorungroj.

**Project administration:** Teerapat Nualnoi, Supawadee Naorungroj.

**Resources:** Supawadee Naorungroj.

**Supervision:** Supawadee Naorungroj.

**Validation:** Teerapat Nualnoi, Supawadee Naorungroj.

**Visualization:** Teerapat Nualnoi, Supawadee Naorungroj.

**Writing – original draft:** Teerapat Nualnoi.

**Writing – review & editing:** Teerapat Nualnoi, Luelak Lomlim, Supawadee Naorungroj.

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
