## [Decision Letter · Decision Letter 0]

11 Mar 2024

Dear Dr. Nualnoi,

Thank you very much for submitting your manuscript "Accuracy of rapid lateral flow immunoassays for human leptospirosis diagnosis: a systematic review and meta-analysis" for consideration at PLOS Neglected Tropical Diseases. As with all papers reviewed by the journal, your manuscript was reviewed by members of the editorial board and by several independent reviewers. The reviewers appreciated the attention to an important topic. Based on the reviews, we are likely to accept this manuscript for publication, providing that you modify the manuscript according to the review recommendations. 

Both have suggested minor revisions to be made. Both reviewers have noted that there is significant heterogeneity among the tests results. The authors should address these points in their revisions accordingly.

Sincerely,

Manisha Biswal

Academic Editor

Ana LTO Nascimento

Section Editor

Both have suggested minor revisions to be made. Both reviewers have noted that there is significant heterogeneity among the tests results. The authors should address these points in their revisions accordingly.

Reviewer's Responses to Questions

**Key Review Criteria Required for Acceptance?**

**Methods**

-Are the objectives of the study clearly articulated with a clear testable hypothesis stated?

-Is the study design appropriate to address the stated objectives?

-Is the population clearly described and appropriate for the hypothesis being tested?

-Is the sample size sufficient to ensure adequate power to address the hypothesis being tested?

-Were correct statistical analysis used to support conclusions?

-Are there concerns about ethical or regulatory requirements being met?

Reviewer #1: The objectives of the study are clearly stated, and the study design is appropriate. Databases searched were described, but I miss the "keywords" and specific dates of the search. Selection criteria are there and correct statistical analyses were used to draw conclusions.

Reviewer #2: The objective is clear, and the study design is appropriate, authors were strict in adhering to guidelines and in the review of the selected studies. However, not including articles in Spanish and Portuguese (among other languages) might limit the results as regions that speak these languages produce a lot of research on leptospirosis.

**Results**

-Does the analysis presented match the analysis plan?

-Are the results clearly and completely presented?

-Are the figures (Tables, Images) of sufficient quality for clarity?

Reviewer #1: Results are clearly described and match with the methods. Results of the quality assessment are assessed and presented in good quality and understable way.

Reviewer #2: The results are clear and adequate for the type of study, figures, and tables are sufficient and clear, and well complimented with the supplements. However, in your Forest Plot analysis, you show an I2 value of 95.76 for sensitivity and 96.35 for specificity which suggests that the studies included might have been too heterogeneous for a metanalysis.

**Conclusions**

-Are the conclusions supported by the data presented?

-Are the limitations of analysis clearly described?

-Do the authors discuss how these data can be helpful to advance our understanding of the topic under study?

-Is public health relevance addressed?

Reviewer #1: Conclusions are supported by the data presented. It is not easy to draw conclusions on the sensitivity and specificity of leptospirosis diagnostic LFIs, when there is so much heterogeneity among the tests. This has been explained very well, together with limitations and considerations. Final conclusion in the last paragraph could be a bit more strong.

Reviewer #2: There is no conclusion section, some conclusions are presented in the discussion but a section with the author's main conclusions presented in a clear and summarized way is needed.

Public health relevance of the study results is not well addressed, the authors could take a stand on whether they believe current evidence supports the use of LFIs in clinical practice (exp: useful, not useful, or not enough evidence to determine their usefulness).

**Editorial and Data Presentation Modifications?**

Reviewer #1: No editorial comments.

Reviewer #2: I suggest it be accepted after minor modifications.

Line 1: The name registered in PROPERO for the register of the systematic review is different from the name used in this paper (In PROSPERO the authors use the word “Performance”, here they use the word “Accuracy”)

Line 63: Clarify your idea, transmission occurs through direct contact with fluids from infected animals, not the animal itself.

Lines 65-64: Fever might be the most common symptom among patients who seek medical care, but most patients present asymptomatic of interspecific symptoms.

Line 73: Replace “in the genus of” by belonging to the genus

Line 108: IFA is not considered a reference test for the diagnosis of leptospirosis, only MAT, culture, or PCR. However, it can be taken into account because this test generates a quantitative result of 1:400 or a fourfold increase in the control sample. The authors made reference to the reason they included studies that used IFA as a reference test in the discussion (lines 304-306), however, they might consider making this clear in the methodology.

Line 159: Review the grammar of that sentence. (remove “for reasons”)

Line 173: Review the grammar of that sentence. (remove the s from “numbers”, remove the “were”, and remove the “per study” (the variation in sample sizes was among the different studies not within each study))

Lines 184-192: At the start of the paragraph when you cite the number of studies you include the percentage they represent in your sample (lines 184 and 185) however after that you just cite the number of studies without providing the percentage they represent (lines 185, 187, 188, and 192). Use one or the other but try to keep the same format.

Line 379: 20 represents the number of LFIs used by the different studies included in your findings. However, can you state this corresponds to the number of developed and marketed LFIs (remember that you only included studies in English)?

**Summary and General Comments**

Reviewer #1: Significant, well-written study, that tries to fill a clinical gap about the accuracy of rapid diagnostic tests for human leptospirosis. Although the heterogenicity of the tests limits the authors to give some clinical relevant directions and novelty, the authors were able to formulate the answers on their objectives.

Reviewer #2: The study was clear and well done, the subject is important because once again. The findings point to the fact that rapid tests (LFIs) do not have an adequate performance for the diagnosis of leptospirosis and the reference tests (MAT, PCR, and culture) are difficult to implement and apply in areas where the disease prevails, which leads to underdiagnosis, under-registration and inadequate treatment.

PLOS authors have the option to publish the peer review history of their article (what does this mean?). If published, this will include your full peer review and any attached files.

Reviewer #1: No

Reviewer #2: No

Figure Files:

Data Requirements:

Reproducibility:

References

---

## [Editor Report · Decision Letter 1]

16 Apr 2024

Dear Dr. Nualnoi,

Thank you very much for submitting your manuscript "Accuracy of rapid lateral flow immunoassays for human leptospirosis diagnosis: a systematic review and meta-analysis" for consideration at PLOS Neglected Tropical Diseases. As with all papers reviewed by the journal, your manuscript was reviewed by members of the editorial board and by several independent reviewers. The reviewers appreciated the attention to an important topic. Based on the reviews, we are likely to accept this manuscript for publication, providing that you modify the manuscript according to the review recommendations. 

Dear Authors, 

 The revised version was sent to two reviewers who feel it that some minor changes are still required. Please go through their comments and revise as appropriate.

Sincerely,

Manisha Biswal

Academic Editor

Ana LTO Nascimento

Section Editor

Dear Authors, 

 The revised version was sent to two reviewers who feel it that some minor modifications are required. Please go through their comments and revise as appropriate.

Figure Files:

Data Requirements:

Reproducibility:

References

---

## [Editor Report · Decision Letter 2]

29 Apr 2024

Dear Dr. Nualnoi,

We are pleased to inform you that your manuscript 'Accuracy of rapid lateral flow immunoassays for human leptospirosis diagnosis: a systematic review and meta-analysis' has been provisionally accepted for publication in PLOS Neglected Tropical Diseases.

Best regards,

Manisha Biswal

Academic Editor

Ana LTO Nascimento

Section Editor

---

## [Editor Report · Acceptance letter]

8 May 2024

Dear Dr Nualnoi,

We are delighted to inform you that your manuscript, "Accuracy of rapid lateral flow immunoassays for human leptospirosis diagnosis: a systematic review and meta-analysis," has been formally accepted for publication in PLOS Neglected Tropical Diseases.

Best regards,

Shaden Kamhawi

co-Editor-in-Chief

Paul Brindley

co-Editor-in-Chief
